

# A novel biomechanical analysis of gait changes in the MPTP mouse model of Parkinson's disease

Werner J. Geldenhuys[1], Tamara L. Guseman[2], Ilse S. Pienaar[3], Dean E. Dluzen[2,4] and Jesse W. Young[2]

[1] Department of Pharmaceutical Sciences, College of Pharmacy, Northeast Ohio Medical University (NEOMED), Rootstown, OH, USA
[2] Department of Anatomy and Neurobiology, College of Medicine, Northeast Ohio Medical University, (NEOMED), Rootstown, OH, USA
[3] Center for Neurodegeneration and Neuroinflammation, Division of Brain Sciences, Department of Medicine, Imperial College London, London, United Kingdom
[4] Current affiliation: Department of Anatomy, Southern Illinois University School of Medicine, Carbondale, IL, USA

Corresponding author
Jesse W. Young,
jwyoung@neomed.edu

## ABSTRACT

Parkinson's disease (PD) is an age-associated neurodegenerative disorder hallmarked by a loss of mesencephalic dopaminergic neurons. Accurate recapitulation of the PD movement phenotype in animal models of the disease is critical for understanding disease etiology and developing novel therapeutic treatments. However, most existing behavioral assays currently applied to such animal models fail to adequately detect and subsequently quantify the subtle changes associated with the progressive stages of PD. In this study, we used a video-based analysis system to develop and validate a novel protocol for tracking locomotor performance in the 1-methyl-4-phenyl-1,2,3,6-tetrahydropyridine (MPTP) mouse model of PD. We anticipated that (1) treated mice should use slower, shorter, and less frequent strides and (2) that gait deficits should monotonically increase following MPTP administration, as the effects of neurodegeneration become manifest. Video-based biomechanical analyses, utilizing behavioral measures motivated by the comparative biomechanics literature, were used to quantify gait dynamics over a seven-day period following MPTP treatment. Analyses revealed shuffling behaviors consistent with the gait symptoms of advanced PD in humans. Here we also document dramatic gender-based differences in locomotor performance during the progression of the MPTP-induced lesion, despite male and female mice showing similar losses of striatal dopaminergic cells following MPTP administration. Whereas female mice appeared to be protected against gait deficits, males showed multiple changes in gait kinematics, consistent with the loss of locomotor agility and stability. Overall, these data show that the novel video analysis protocol presented here is a robust method capable of detecting subtle changes in gait biomechanics in a mouse model of PD. Our findings indicate that this method is a useful means by which to easily and economically screen preclinical therapeutic compounds for protecting against or reversing neuropathology associated with PD neurodegeneration.

## INTRODUCTION

Parkinson's disease (PD) is an age-related neurodegenerative disease, where individuals aged older than 60 years of age show increased risk of developing the disorder (*Connolly & Lang, 2014*). A triad of classical motor symptoms is seen in advanced PD patients, consisting of rigidity, akinesia, and tremor (*DeLong & Wichmann, 2009*). These symptoms appear following the loss of at least 80% of the dopaminergic neurons within the Substantia Nigra pars compacta (SNpc) (*Hartmann, 2004*), thereby impairing a patient's ability to perform everyday tasks (*Aviles-Olmos et al., 2013*). As the disease progresses, co-morbid non-motor symptoms manifest, including cognitive impairments and depression (*Lawson et al., 2014*; *Obeso et al., 2014*), which are often resistant to dopamine (DA) replacement therapies (*Connolly & Lang, 2014*). Currently available therapies are mainly aimed at replacing the lost striatal DA content. With dopaminergic cell loss continuing and the side-effects associated with synthetic DA replacement increasing, continuous use of pharmacotherapeutics fails to alter disease progression (*Connolly & Lang, 2014*). On the other hand, deep brain stimulation (DBS) therapy shows dramatic improvements in some late-stage PD patients, including improved gait and postural instability following DBS implanted in the pedunculopontine nucleus (PPN), arguing strongly for a case that patients might benefit substantially more, should intervention be initiated at an earlier stage of the disease progression (*Mazzone et al., 2014*). Related to this, recent work revealed that DBS targeting the subthalamic nucleus (STN) induces vascular remodeling effects, including an upregulation of the vascular endothelial growth factor (VEGF), suggesting that DBS induces plasticity-related effects (*Pienaar et al., 2015*). Hence, optimization of intervention protocols stand to benefit greatly from a reliable animal model of PD that mimics progressive stages of the disease, in line with the clinical aim of initiating treatment at an earlier stage during progressive PD.

Accurate recapitulation of the movement phenotype seen in PD patients in animals is important for the assessment and development of novel therapeutic treatments as well as for providing a tool by which to gain insights into the molecular and cellular mechanisms contributing to the loss of neurons and the concomitant circuit disruptions that characterize human PD. In this regard, sensitive behavioral paradigms are of paramount importance for characterizing existing and newly introduced genetic-based and toxin-induced animal models of neurodegenerative disease (*Bury & Pienaar, 2013*; *Pienaar, Lu & Schallert, 2012*). Unfortunately, many of the behavioral assays applied to animal models fail to adequately detect the subtle changes associated with the different stages of the disease (*Antony, Diederich & Balling, 2011*; *Meredith & Rademacher, 2011*). In this study, we used a novel video-based paradigm for analyzing gait and locomotor kinematics to detect the subtle longitudinal changes in locomotor performance occurring in the methyl-4-phenyl-1,2,3,6-tetrahydropyridine (MPTP) mouse model of PD, and analyzed the results in a gender-specific manner. Mice and non-human primates systemically injected with MPTP, a mitochondrial neurotoxin, show loss of mesencephalic dopaminergic neurons with concomitant loss of striatal DA content, resulting in motor deficits (*Bezard & Przedborski, 2011*; *Pienaar, Lu & Schallert, 2012*;

*Schmidt & Ferger, 2001*). We anticipated that (1) MPTP-treated mice should move at slower speeds and with shorter, less frequent strides, mimicking the bradykinesia shown by PD patients (*Fernagut et al., 2002*) and that (2) gait deficits should monotonically increase following MPTP administration, as the effects of neurodegeneration become manifest (*Klemann et al., 2015*). Finally, we also tested for a gender effect in the locomotor response to MPTP treatment, given that previous research has established sex-based differences in the phenotype of both PD patients and rodent models of the disease (*Antzoulatos et al., 2010*; *Gillies et al., 2014*; *Van Den Eeden et al., 2003*).

## MATERIALS AND METHODS

### Study design

Research was carried out at Northeast Ohio Medical University (NEOMED), in strict accordance with the recommendations in the Guide for the Care and Use of Laboratory Animals of the National Institutes of Health. All procedures were pre-approved by the NEOMED Institutional Animal Care and Use Committee (NEOMED IACUC Protocol 10-006).

Three cohorts of C57BL/6J laboratory mice ($n = 4$ per group, each consisting of 2 males and 2 females) were used. One male mouse from the first cohort and one female mouse from the second cohort did not show the characteristic striatal DA depletion following MPTP treatment. Additionally, one female mouse from the third cohort lost a significant amount of body weight following MPTP treatment and was removed from the study. Therefore, our final sample consisted of nine mice (five males and four females). On average, male mice weighed more than the female mice (mean female body mass [95% confidence limits]: 23.7 g [23.02 g, 24.32 g], mean male body mass: 29.1 g [27.56 g, 30.66 g]; measured prior to MPTP treatment).

The three cohorts were tested in chronological order, such that at the start of data collection, the mice in cohort 1 were 12 weeks of age, those in cohort 2 were 14 weeks of age, and those in cohort 3 were 16 weeks of age. Following a training period lasting from 3 to 4 days, during which mice were acclimated to the experimental procedure, each animal was evaluated in an initial behavioral assessment to quantify baseline locomotor performance. Following this baseline assessment, mice were treated with a single dose of MPTP (35 mg/kg; i.p.) and then evaluated for seven days of longitudinal locomotor testing, starting at the first day post treatment, but at least 24 h after MPTP treatment to bypass some of the most acute phenotypic effects due to the drug's toxicity (e.g., epilepsy-like symptoms: *Klemann et al., 2015*). In addition to the baseline evaluation, every mouse in the dataset was evaluated at the seventh day after treatment. However, to improve the efficiency of our data collection and reduce the activity burden on the mice, on the remaining days (i.e., days 1–6 following MPTP treatment) experiments were staggered to ensure that we collected data from at least two cohorts of mice per experimental day (i.e., cohort 1 was evaluated on days 1–4, cohort 2 on days 1, 2, 5, 6, and cohort 3 on days 3–6). Mice were then euthanized via cervical dislocation, the striata removed and the DA levels determined using high-pressure liquid chromatography (HPLC), described below.

We chose not to include a "sham" control group in our experimental design, but to rather use mixed-model repeated measures statistical analyses (detailed below) to test for the gait-related sequelae of MPTP treatment against each individual's pre-treatment baseline. This design reduced the number of animals required for our study, satisfying ethical concerns relating to reducing the number of animal subjects used for achieving robust statistical results. Our statistical design was combined with an intensive data collection protocol (15–25 locomotor trials collected per day, per experimental animal), enabling us to accurately characterize the efficacy of our new method and obtain statistically significant results, despite a relatively small sample size.

## Apparatus and locomotor testing

The mice were made to run along a white wooden trackway (41.75 cm long, 4 cm wide) into a dark box located at the terminal end of the track. The trackway was placed on a laboratory bench top, approximately 1 m off the ground. Mice were filmed with a high-speed digital camera (MotionScope Model PCI 1000s; Redlake MASD Inc., San Diego, California, USA) placed overhead to allow for a dorsal view of the mouse running along the trackway. Videos were recorded at 200 frames per second (fps) with a 1/2,000 shutter speed. The trackway was illuminated with a 250-watt quartz light (Lowel-Light Manufacturing Inc., Brooklyn, New York, USA) to provide adequate depth of field. On either side of the runway, mirrors were placed at 45° angles to the sagittal plane to provide complete views of the footfalls of each limb during locomotion (Fig. 1). We recorded 15–25 strides per mouse per day. All testing took place between 10AM and 2PM, during the active (dark) phase of these nocturnal animals.

## Analyses of locomotor performance

The recorded sequences were ported into ProAnalyst motion analysis software (Xcitex Inc., Woburn, Massachusetts, USA) for off-line analyses. We marked the touchdown and lift-off of each forelimb and hindlimb, recognizing touchdown as the first frame in which the limb contacted the runway and lift-off as the first subsequent frame in which the limb was no longer in contact. Individual locomotor strides were identified based on the touchdown of a reference limb (e.g., from a touchdown of the left hindlimb to the next touchdown of the same limb). Only "symmetrical" walking and running strides were included in the final dataset, excluding all high-speed "asymmetrical" bounding and galloping strides (*Hildebrand, 1965*). Strides during which a mouse walked on the side mirrors were excluded from the dataset. Overall, approximately 80% of recorded strides were included in the final dataset.

We digitized the two-dimensional position of the tip of the nose and the base of the tail across each pass down the runway. To mitigate digitizing errors and interpolate feature positions across occluded frames, each digitized feature was fit to a quintic smoothing spline function (tolerance: 1 mm$^2$) using MATLAB (version R2014b; MathWorks, Natick, Massachusetts, USA) (*Walker, 1998*). In order to measure an individual animal's net movement trajectory, we modeled mice as point masses localized at the average static center of mass (COM) position for each gender. Measurements of static COM position in

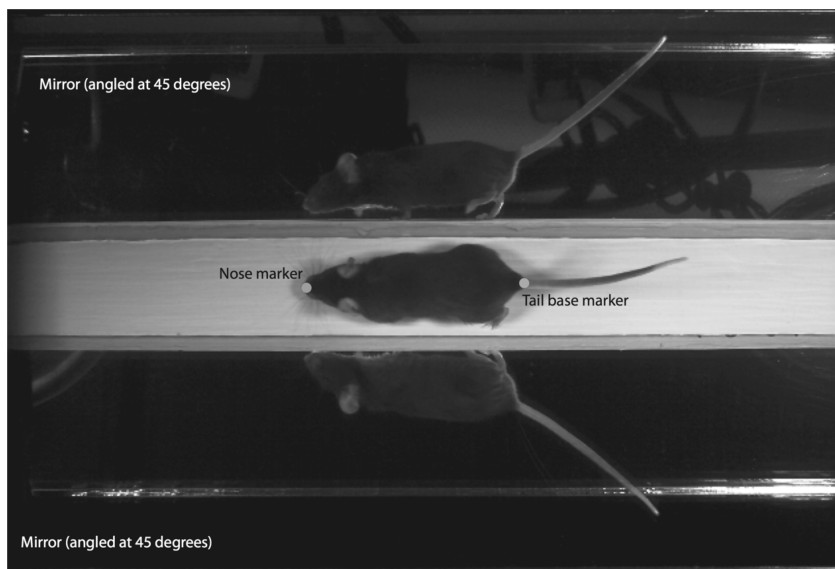

**Figure 1 Sample frame from high-speed video analyses of mouse locomotion.** Video analysis software was used to digitize the two-dimensional position of the nose and tail base during the pass down the trackway. A pair of mirrors angled at 45° to the sagittal plane allowed for capturing touchdown and lift-off events over complete gait cycles.

cadaveric male and female mice ($N = 3$ males and $N = 2$ females) using the reaction board method (*Ozkaya & Nordin, 1999*; *Young, 2012*) showed some variation between genders, but little variation between individuals within genders (male mean COM position: 41.5% of nose-to-tail base length, coefficient of variation: 6.9%; female mean COM position 45.5% of nose to tail length, coefficient of variation 3.1%). The COM was therefore defined as a point 41.5% along the length of the nose-to-tail base vector in males and 45.5% along the length of the nose-to-tail base vector in females.

Although our proxy is necessarily a simplification of actual COM position, which can be expected to vary based on both body configuration during locomotion, variation in the effective length of the trunk (i.e., the euclidean distance between the nose and the tail base) was minimal during locomotor strides (average coefficient of variation with 95% confidence bounds, 4.08% [3.822%, 4.337%] in males, 3.81% [3.507%, 4.114%] in females). These data suggest that little lateral flexion of the vertebral column had occurred and indicate that our approach of modeling the trunk as a linear vector should have minimal effect on the accuracy of our COM estimates.

Raw data on footfall timings and modeled COM displacements were used to calculate several metrics of locomotor performance. Average locomotor speed (cm/s) was calculated as the absolute value of mean velocity across the stride. Stride length (cm) was defined as the net distance travelled by the COM during the stride. Stride frequency (Hz) was calculated as the inverse of stride duration. Additionally, we calculated two metrics of overall postural stability. First, based on the timings of limb touchdown and liftoff events, we calculated the percentage of stride duration spent in various support combinations

(e.g., supported by one, two, three or four limbs). We then calculated the mean support number using the equation: % single-limb support + (2 × % double-limb support) + (3 × % triple-limb support) + (4 × % quadruple-limb support). A higher mean support number indicates that a greater number of limbs provide support at any one instance during the stride, theoretically conferring greater postural stability to the animal. Additionally, we calculated a "sway" index to quantify mediolateral stability, based on the "straightness" index of *Jamon & Clarac (1998)*. The sway index was computed as the ratio of the total horizontal path distance travelled by the COM during the stride and the straight-line distance between the COM coordinates at the beginning and the end of the stride, multiplied by 100. A sway index of 100 indicates a perfectly linear path with higher values indicating increasing amounts of mediolateral sway.

## DA content analysis

We measured remaining levels of DA following systemic MPTP treatment, as previously described (*Geldenhuys et al., 2014*). In brief, mice were euthanized via cervical dislocation, the brain was removed from the skull, and striatal tissue was dissected out then snap frozen in liquid nitrogen. For processing, the striatal tissue was weighed and placed in cold perchloric acid (0.1 N, 500 µL, 4 °C). Tissue samples were sonicated and centrifuged. An aliquot was removed to measure DA levels within the bilateral striata across the various cohorts. Tissue samples were evaluated for DA content by means of HPLC coupled with electrochemical detection. Biogenic amines were separated on a Supelco column (Discovery C18, 10 cm × 3 mm × 5 µm). Samples were injected into a 20 µL loop. A degassed isogradient mobile phase consisting of sodium acetate (50 mM), citric acid (27.4 mM), sodium hydroxide (10 mM), sodium octyl sulfate (0.1 mM), ethylenediaminetetraacetic acid (EDTA) (0.1 mM) and 5% methanol in filtered deionized water was used for the system. The mobile phase was adjusted to a final pH of 4.5 with the addition of NaOH, with was and filtered (0.45 µm, Millipore filter; Millipore, Billerica, Massachusetts, USA) prior to use. Standards were diluted in perchloric acid (0.1 N) in increments of 3.1, 6.2, 12.5, 25, 50, 100, 200, and 400 pg/20 µL. Samples were analyzed by using the Chromelian 6.8 software program (Dionex, Sunnyvale, California, USA). The assay sensitivity (6.2–12.5 pg/20 µL) was determined by observing reliable peaks above baseline noise. Striatal DA values were compared to DA levels in a vehicle-control group of six mice (three males and three females) made up of littermates from the three cohorts of MPTP-treated mice (i.e., one male and one female from each cohort).

## Statistical analysis

We used a rank-based Student's *t*-test with the Welch–Satthewaite correction for heteroscedasticity to compare DA loss between genders. This method was recommended by *Ruxton (2006)* as the most robust method for two-group mean comparisons. We used linear mixed-effects Analyses of Covariance (ANCOVA) to test for gender differences in the longitudinal effects of MPTP toxicity on locomotor performance. The mixed-effects model allowed us to incorporate all relevant fixed factors into our analyses (i.e., gender and days since MPTP treatment), whilst controlling for random variation amongst the

**Table 1** Sample sizes of locomotor strides, grouped by gender and MPTP treatment day.

| | Baseline | MPTP treatment day | | | | | | |
| | | Day 1 | Day 2 | Day 3 | Day 4 | Day 5 | Day 6 | Day 7 |
|---|---|---|---|---|---|---|---|---|
| Female | 56 | 40 | 28 | 23 | 24 | 39 | 35 | 42 |
| Male | 68 | 43 | 30 | 29 | 39 | 53 | 56 | 64 |

mice within each gender group (*Batka et al., 2014*). All measures were averaged across strides within each experimental day for each mouse, so that the unit of analysis was set as the average response of each mouse on each day of testing. We fitted a full model for each measure, testing for (1) a main effect of gender, (2) a significant association with the number of days since MPTP treatment and (3) an interaction between gender and the number of days since MPTP treatment. A significant interaction indicates that males and females responded differently to MPTP toxicity. In this case, we report separate gender-specific regression slopes (and 95% confidence intervals on those slopes) in order to better illustrate the magnitude and direction of the interaction effect. Additionally, due to the pervasive influence of locomotor speed on most measures of rodent locomotor performance (*Batka et al., 2014*), in the event of a significant gender-specific regression, we reanalyzed the relationship between the performance variable and the days since MPTP treatment with speed included as a covariate in the model. In these cases, any significant residual association between a performance variable and the days since MPTP treatment indicates that the indicated change in gait mechanics is not simply an after-effect of mice changing their average locomotor speed. All statistical analyses were performed using the R statistical package (*R Core Team, 2013*), including the nlme (*Pinheiro et al., 2013*) and lsmeans (*Lenth, 2014*) add-on packages.

# RESULTS

## Striatal DA levels

Relative to control values, striatal DA levels in MPTP-treated mice decreased by an average of 54.8% amongst treated males and 58.8% amongst treated females (Fig. 2). Although striatal DA values in both genders were significantly lower than those in the vehicle control group (males: $t_{[4]} = -7.8, p = 0.001$; female: $t_{[3]} = -7.7, p = 0.005$), loss of striatal DA was statistically similar between genders ($t_{[7]} = 0.48, p = 0.64$).

## Locomotor performance

A total of 668 symmetrical strides were analyzed for this study (287 strides from female mice, 381 strides from male mice). A breakdown of the number of strides coded for each experiment day and grouped by gender is provided in Table 1. Variation in the number of valid strides available for analysis on each day accounts for the unequal number of strides across the tabular cells.

Mixed-effects analyses of covariance indicated significant gender-by-days-since-MPTP-treatment interaction for all variables ($p \leq 0.029$; results summarized in Table 2). In each

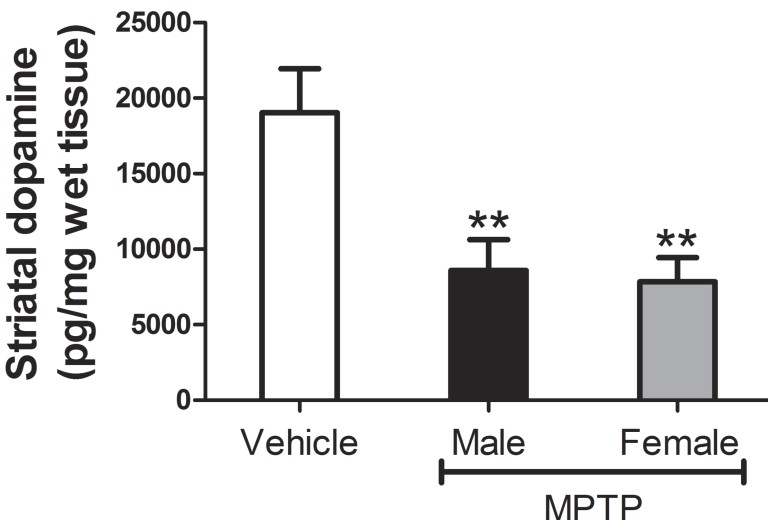

**Figure 2 Measures of striatal DA levels in male and female mice.** Each bar represents mean $\pm$ S.D. ($N$ females $= 4$, $N$ males $= 5$). Asterisks indicate that DA levels in treated males and females were significantly lower than in the vehicle-control group ($p < 0.01$ for both genders).

**Table 2 Analyses of covariance of locomotor performance variables by gender and MPTP treatment day.**

| Variable | Interaction effect[a] | Slope [95% confidence interval][b] |
|---|---|---|
| Speed | $-1.87$ $F_{[1,43]} = 7.93$ $p = 0.007$ | **Female slope:** 0.616 [$-0.373$, 1.61] **Male slope:** $-1.25$ [$-2.15$, $-0.351$] |
| Stride length | $-0.0998$ $F_{[1,43]} = 7.69$ $p = 0.008$ | **Female slope:** 0.0365 [$-0.0173$, 0.0902] **Male slope:** $-0.0633$ [$-0.112$, $-0.0145$] |
| Stride frequency | $-0.136$ $F_{[1,43]} = 7.05$ $p = 0.011$ | **Female slope:** 0.0412 [$-0.0351$, 0.118] **Male slope:** $-0.0945$ [$-0.164$, $-0.0252$] |
| Mean support number | 0.0254 $F_{[1,43]} = 5.07$ $p = 0.030$ | **Female slope:** 0.00355 [$-0.0133$, 0.0204] **Male slope:** 0.0290 [0.0136, 0.0443] |
| Sway index | $-0.0539$ $F_{[1,43]} = 8.5$ $p = 0.006$ | **Female slope:** 0.00996 [$-0.0176$, 0.0375] **Male slope:** $-0.0439$ [$-0.0689$, $-0.0189$] |

**Notes.**

[a] The interaction effect indicates the average difference in slope between male and female mice (i.e., male slope–female slope), and the significance of this difference relative to the null expectation of zero. Significant interactions are indicated by a $p$-value in printed in **bold type**.

[b] These values indicate the gender specific slope of the performance variable against MPTP treatment day. 95% confidence intervals are displayed as [lower confidence bound, upper confidence bound]. Confidence bounds of opposite sign indicate a non-significant slope.

case, post-hoc analysis of gender-specific regression slopes indicated that the locomotor performance of males significantly changed following MPTP administration, whereas females showed no significant response to MPTP treatment (i.e., the 95% confidence

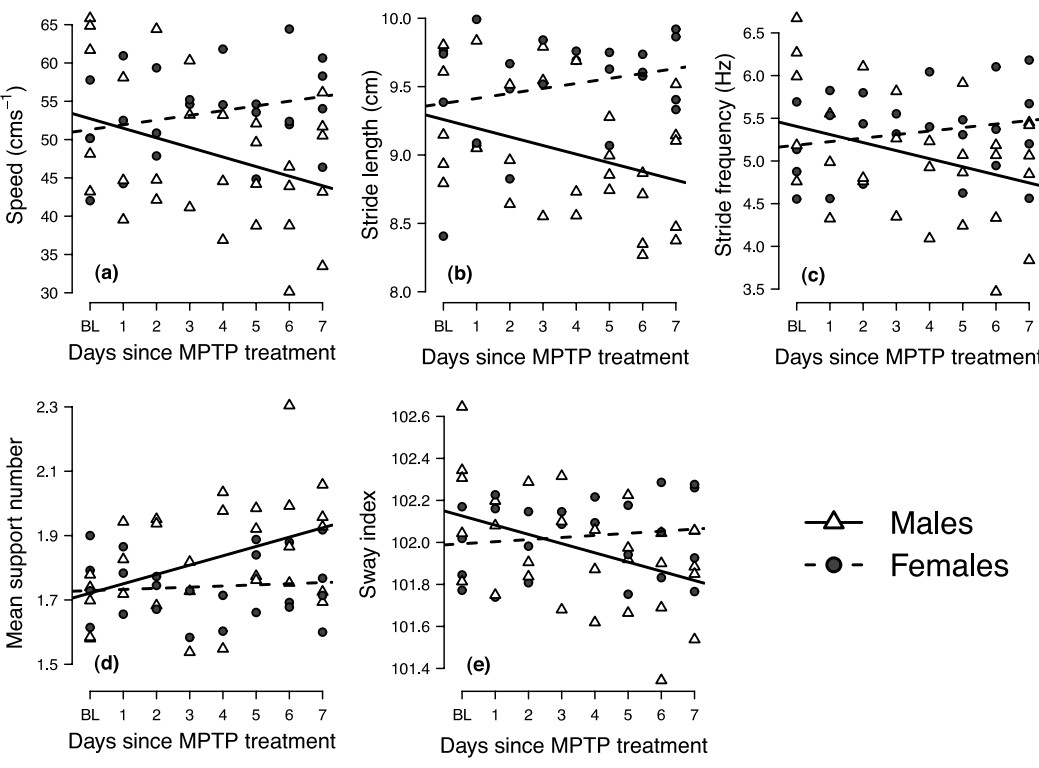

**Figure 3** **Longitudinal changes in locomotor performance following MPTP administration in male and female mice.** Panels show the individual mean values of (A) speed, (B) stride length, (C) stride frequency, (D) mean support number and (E) sway index at baseline (BL) at each day following MPTP treatment. Trend lines indicate gender-specific linear mixed-effects regression slopes.

intervals about the female slopes encompassed zero). Male mice responded to MPTP toxicity by decreasing speed, taking shorter and fewer strides (i.e., decreasing stride length and frequency), increasing the average number of supporting limbs on the ground during the stride (i.e., increasing mean support number) and decreasing mediolateral sway (i.e., lower sway indices) (Table 2 and Fig. 3).

Because speed has been shown to exert a pervasive influence on other measures of locomotor performance in rodents (*Batka et al., 2014*), we reassessed the significant relationships between the number of days following MPTP treatment and stride length, stride frequency, mean support number, and sway index in the male mice, whilst controlling for speed using linear mixed-effects multiple regression (results summarized in Table 3). All four variables were significantly associated with speed ($p < 0.001$), suggesting that longitudinal changes in speed may have affected other measures of locomotor performance. Indeed, after controlling for speed, longitudinal decreases in stride length and stride frequency were found to no longer reach statistical significance ($p \geq 0.415$). In contrast, longitudinal changes in mean support number and sway index remained significant even after including speed as a covariate in the model (all $p \leq 0.032$).

**Table 3** Association between locomotor performance variables and MPTP treatment day, controlling for speed.

| Variable | Partial regression coefficient [95% confidence interval][a] |
|---|---|
| Stride length | −0.016 [−0.0554, 0.0237] $p = 0.415$ |
| Stride frequency | −0.002 [−0.0254, 0.0209] $p = 0.842$ |
| Mean support number | 0.010 [0.0001, 0.0196] $p = \mathbf{0.032}$ |
| Sway index | −0.012 [−0.0214, −0.0018] $p = \mathbf{0.022}$ |

**Notes.**
[a] The partial regression coefficients reflect the residual influence of MPTP treatment day on each locomotor performance variable, controlling for the effects of speed. Significant partial regressions are indicated by a $p$-value in printed in **bold type**.

## DISCUSSION

Tests designed to economically and accurately assess gait function in PD animal models are currently lacking (but see *Amende et al., 2005*). Sensitive assessment tools capable of capturing the different stages of PD in well-established animal models of the human condition remain a challenge to PD researchers interested in screening and developing treatments and diagnostic markers for translational use (*Pienaar, Lu & Schallert, 2012*). Here we show that videography-based measurements of gait kinematics represent a robust method to assess behavioral deficits in a well-validated mouse model of PD. In PD patients, alterations in the functioning of the nigrostriatal dopaminergic system (NSDA) manifest as changes in sensorimotor functions (*DeLong & Wichmann, 2009*). In mice, PD can be modeled to a remarkably accurate extent with systematic treatment of the relatively selective NSDA neurotoxin, MPTP. A potential drawback of this model is that seeing early changes in the motor behavior of mice when treated with MPTP may be difficult to observe (*Blandini & Armentero, 2012*; *Bove & Perier, 2012*; *Morin, Jourdain & Di Paolo, 2014*; *Tieu, 2011*). One possible explanation for this difficulty in detecting early motor changes in mice is that current behavioral assays employed by PD researchers are insensitive to the subtle behavioral changes shown by genetic-based or toxin-induced animal models of human PD (*Pienaar, Lu & Schallert, 2012*). These metrics, which typically include gait measures (e.g., stride length: *Fernagut et al., 2002*), the pole test (i.e., time required to descend a given pole: *Matsuura et al., 1997*) for assessing bradykinesia, performance on a rotarod spindle for measuring balance, grip strength and motor coordination (*Rozas, Guerra & Labandeira-García, 1997*), and a balancing test assessed by means of a beam walking task (*McDermott et al., 1994*). However, these tests provide only a gross assessment of sensorimotor function, with little distinction made between the precise functional impairments. Additionally, in many of these assays, rodents are required to perform tasks that are outside of their natural locomotor behavior, making it difficult to see subtle changes in function.

Accordingly, marked benefits could ensue with the use of novel approaches and perspectives for the study of motor behavior in animal models of PD. In particular, detailed analyses of gait and balance during normal rodent locomotion promise to reveal subtle changes in motor behavior that may be difficult to detect using common techniques (*Wang et al., 2012*). Moreover, such analyses have the potential to better model the bradykinesia (slowed movements), festination (short, shuffling steps), postural abnormality, and gait instability that characterizes PD patients. The significance of such an approach is the prospective for identifying specific sensorimotor behavioral markers of NSDA neurodegeneration. Since symptoms associated with PD are not evident until advanced stages of NSDA neurodegeneration, the possibility for identifying behavioral changes at an earlier stage of lesion progression in animal models of PD may permit more accurate screening of novel drugs or stimulatory implants, informing our understanding of the etiology of the human form of the disease and leading to more promising treatments.

One approach for performing such sophisticated analyses of motor behavior consists of the biomechanical analysis of gait dynamics. Recently, Wang and others (*2012*) commented that gait measures specifically focusing on murine models of PD are currently lacking in the literature. The precise, quantitative descriptions of animal movement presented in this study have the potential to identify more subtle behavioral outputs of motor-related neurodegeneration than currently available locomotor assays offer. Moreover, the robust comparative literature that currently exists on the biomechanics of animal quadrupedal gait (*Batka et al., 2014*; *Young, 2012*) offers a solid methodological and theoretical framework from which to construct and evaluate the analyses used in our research.

The use of the video analysis system was able to give significant insights into the gait deficits, which were subtle at times, of MPTP-induced Parkinsonian mice. In male mice, but not in the females, there was a significant change in gait behavior following the loss of striatal DA content (Fig. 3). Using our novel experimental protocol, we were able to observe changes in the gait dynamics of the mice soon after MPTP administration that correlated with the bradykinesia and hyper-rigidity seen in PD patients (*Hanakawa et al., 1999*). Specifically, following MPTP administration, male mice used slower, shorter and less frequent strides, selected footfall patterns that ensured a greater number of supporting limbs across the stride and decreased mediolateral sway (Fig. 3). The gait deficits seen in the male mice became more pronounced throughout the period following MPTP treatment, indicating a longitudinal decrease in locomotor performance as the neurodegenerative effects of MPTP-toxicity became more pronounced (*Klemann et al., 2015*). Although loss in stride length and frequency appeared to be side effects of the general decrease in locomotor speed, longitudinal changes in mean support number and sway index took place independently of the changes in locomotor speed (Table 3). The longitudinal increase in mean support number suggests that male mice altered their gait in some way to ensure that a greater number of limbs contacted the floor surface throughout the stride, serving as a compensatory effect for promoting greater gait control, to likely be an attempt at increasing locomotor stability. Similarly, the longitudinal decline in the sway index may indicate a decreased ability to cope with challenges to mediolateral stability, as

recently demonstrated in human PD patients (*Galna, Murphy & Morris, 2013*). Although it is possible that some of the gait changes observed during the first 1–2 days post-injection could have been associated with an acute inflammatory response to MPTP, subsequent deficits in locomotor performance are best explained as sequelae of neurodegeneration, mimicking the phenotypic processes that occur during human PD (*Klemann et al., 2015*), mimicking the phenotypic processes that occur during human PD.

Interestingly, MPTP-treated female mice showed a dramatic difference in response to the MPTP toxicity. Taking all of the results into account, female mice showed a delayed and diminished response to MPTP on gait function, agreeing with findings that premenopausal women are at lower risk of developing Parkinsonism (*Van Den Eeden et al., 2003*), and that the clinical phenotype of PD in women often shows a delayed presentation that is reduced in severity when compared to men (*Gillies et al., 2014*). Our findings are also in agreement with previous studies on the gender bias in MPTP treated mice (*Antzoulatos et al., 2010*).

Although the posthumous processing of striatal tissue indicated similar percent losses of dopaminergic neurons in males and females in response to MPTP, C57BL/6J females typically show substantially greater absolute levels of striatal DA (*Dluzen, McDermott & Liu, 1996*). Accordingly, a critical absolute concentration of DA may remain in females to enable their relatively better gait performance. It is also likely that the observed gender differences in gait dysfunction are due to compensatory processes unique to females. Specifically, estradiol has been shown to promote DA turnover, -synthesis, and -release, while simultaneously suppressing reuptake, providing a sex-specific buffering mechanism for preserving behavioral function despite the depletion of dopaminergic populations of neurons (*Dluzen & Horstink, 2003*; *Gillies et al., 2014*). While estrogen is capable of producing neuroprotection against MPTP in both male and female C57BL/6J mice (*Dluzen, McDermott & Liu, 1996*), the greater endogenous levels of estrogen along with greater basal striatal DA concentrations of females may contribute to such compensatory processes resulting in a maintenance of female performance in our locomotor task.

## CONCLUSIONS

Economical tests designed to accurately assess natural gait function in PD animal models are currently lacking. Here we illustrate the ability of a novel testing protocol using video-based analysis of movement to provide insight into the gait performance of MPTP-treated mice. We were able to show changes in gait function in early stages following MPTP treatment. The male group showed a statistically significant higher propensity towards gait changes than the female mice, suggesting that gait deficits in female MPTP-treated mice might be subtler. Future work should consider carrying out the described gait analyses in a cross-sectional manner, allowing for repeated assays of striatal DA content following MPTP administration to facilitate more precise testing for possible associations between striatal DA depletion and locomotor dysfunction. Overall, the results of our study demonstrate clearly that the novel methodology proposed here has the potential to precisely quantify gait changes in animal models of PD. As such, our

measures could be used as output metrics in the initial screening and optimization of compounds and surgical interventions for slowing, or even reversing, disease progression.

## ACKNOWLEDGEMENTS

We thank Bartholomew White and Michael Pante for contributing to data coding and providing technical assistance. The members of the NEOMED Comparative Biomechanics Journal Club provided helpful advice during the preparation of this manuscript.

### Funding

The research was funded by the Department of Pharmaceutical Science and the Department of Anatomy and Neurobiology at Northeast Ohio Medical University (NEOMED). The funders had no role in study design, data collection and analysis, decision to publish, or preparation of the manuscript.

### Grant Disclosures

The following grant information was disclosed by the authors:
Department of Pharmaceutical Science.
Department of Anatomy and Neurobiology at Northeast Ohio Medical University (NEOMED).

### Competing Interests

The authors declare there are no competing interests.

### Author Contributions

- Werner J. Geldenhuys conceived and designed the experiments, performed the experiments, contributed reagents/materials/analysis tools, wrote the paper, prepared figures and/or tables, reviewed drafts of the paper.
- Tamara L. Guseman performed the experiments.
- Ilse S. Pienaar wrote the paper, reviewed drafts of the paper.
- Dean E. Dluzen conceived and designed the experiments, analyzed the data, contributed reagents/materials/analysis tools, wrote the paper, reviewed drafts of the paper.
- Jesse W. Young conceived and designed the experiments, performed the experiments, analyzed the data, wrote the paper, prepared figures and/or tables, reviewed drafts of the paper.

### Animal Ethics

The following information was supplied relating to ethical approvals (i.e., approving body and any reference numbers):

All procedures involving vertebrate animals were approved by the Northeast Ohio Medical University Institutional Animal Care and Use Committee (NEOMED IACUC Protocol 10-006).

## Supplemental Information

Supplemental information for this article can be found online at http://dx.doi.org/10.7717/peerj.1175#supplemental-information.

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
