# Peer review of "A novel biomechanical analysis of gait changes in the MPTP mouse model of Parkinson’s disease"

_PeerJ, doi:10.7717/peerj.1175_

## Round 0.1 · original submission · Major Revisions

· Academic Editor

Major Revisions

Please take care of the suggestions from both reviewers.

·

Basic reporting

The paper is clear, concise and easy to follow. The introduction and background is appropriate. However, I’m not sure if table 1 is really helpful (please see also point 4 below).

Experimental design

The research question in clearly defined, i.e. to “develop and validate a novel protocol for tracking locomotor performance in the 1-methyl-4-phenyl-1,2,3,6-tetrahydropyridine (MPTP) mouse model of PD.” This is indeed a relevant and meaningful topic. The methods – including the statistical aspects – are generally well described.

However, a few questions could be asked here:

1. The authors chose not to use a control group (i.e. no MPTP or sham). Why? This would also solve the issue raised in question 7.

2. The title suggests that the main question addressed concerns the gender difference. However, the abstract and final conclusion suggests that particularly the demonstration of the novel protocol for tracking locomotor performance is key. The authors could perhaps more specifically choose one of either focus, and describe the other one as ‘secondary’ results. Or use the gender differences as illustration/proof of concept to demonstrate the usability of the technique.

3. How was the COM calculated from (only) 2 points? Perhaps a reference could be added? Wouldn’t the difference in body weights between male and female mice (or their body shapes in general) influence this COM (i.e. that there might be a bias due to fixed assumptions about body shape when estimating COM from just a single line in 2D)? And if so, would that have influence on the gender differences?

4. From the figures, it appears that the number of animals included differs from day to day post-injection. This notion is also clear from the supplementary data file. Why were animals excluded on specific days? Did they not show any usable strides that day?

5. Were the six non-treated animals of the same age (range) as the treated mice (as DA levels do change over time in mice).

Validity of the findings

The conclusions are well supported by the results. However, several questions remain.

6. The authors test the mice only up to 7 days post MPTP treatment. How accurately does this represent PD, after a single MPTP dose? It appears that the mice were tested in the (acutely) toxic phase of the treatment. Wouldn’t it have been more accurate to test the animals in a more stable phase? For example, at a molecular level – particularly at striatal levels – acute processes (e.g. linked to epilepsy, not movement disorders) were selectively enriched shortly following MPTP treatment in mice. See Klemann et al, Mol Neurobiol. 2015 Feb 13 [Epub ahead of print].

7. How do the authors separate the motor performance decline from a learning effect (i.e. less anxiety, or less motivation) to account for the slower traverse of the trackway? To avoid this bias, using separate sets of animals to test on each time point after injection would rule out such ‘adaptation bias’ (at the expense of more animals). In fact, this could be combined with striatal dopamine levels over time after MPTP treatment. Another solution would be using a control group (no MPTP/sham).

8. The authors do not elaborate on why there appears to be a gender difference – their main research question. Not only is there a lack of significance in females, there appears to be a trend to opposite effects compared to males. A critical appraisal, perhaps with suggestions for further research, would benefit the discussion.

Reviewer 2 ·

Basic reporting

See comments to the author

Experimental design

See comments to the author

Validity of the findings

See comments to the author

Comments for the author

Geldenhuys et al report that following an acute, single high dose of MPTP, there are differences in several measures of locomotion/gait between male and female mice despite the fact that the loss of striatal dopamine is identical at the end of the 7 day study. The finding of a gender difference in several measures of gait/locomotion is of interest. The authors are to be commended for building a novel apparatus to measure the locomotion, although more expensive and more sensitive gait machines are available on the market.

This reviewer has the following concerns:

1.) Results: Striatal dopamine levels: where is the DOPAC data? It states in the methods section that DOPAC was measured. Present the data. Once you report that data, please calculate the turnover of DA/DOPAC. Also, is the vehicle group a combination of both males and females? This was never described. For Figure 2, the significance of the asterisk was not described.
2.) Results: locomotor performance: In the middle of the second paragraph (since the pages were not numbered, this description will need to suffice). The authors state that the ‘male mice responded to the progression of the MPTP toxicity by decreasing speed, etc’. The acute administration of MPTP is not a model of progression. Such chronic models of progressive DA loss do exist, but such acute treatment is not progressive. Based on the experience of this reviewer, there is significant likelihood that 1 day following a single high dose of MPTP, DA levels have bottomed out and there is no further loss of this neurotransmitter. Since progression implies continued loss of DA, the authors need to measure the levels of DA/DOPAC at each of the time points analyzed for the behavior in order to accurately report that there is a progression of MPTP toxicity. Most likely this is not the case. In addition, motor behavioral was measured starting the next day after acute MPTP. Using this very high acute dose of MPTP, it’s most likely the animals may remain sick for several days. The authors need to address this issue. It is the experience of this reviewer that any locomotor behavior is measured 7 days following the last dose of MPTP in order to rid the body of the MPTP metabolites and any possible sickness due to toxin administration. Measuring behavior within the first week following a high dose of MPTP is not justified.
3.) Results: the other major concern of this reviewer is the incredibly small N for each gender group. An N of 4 or 5 mice, especially when measuring behavior, is far too small a sample size. This reviewer analyzes between 8-12 mice/group when it comes to behavior, essentially double the number of mice the current group used in this study. Once the authors repeat the study with the larger N, that will most likely determine the reproducibility of the behavioral observations.
4.) Discussion: 4th paragraph: In figure 3, the authors claim a correlation between changes in gait and striatal DA content. Where are those data? Please present it.
5.) Discussion: middle of the 4th paragraph: the authors state that progressive locomotor deficits during MPTP treatment is due to the progression of toxicity. The authors present no data showing progressive loss of dopamine. The authors need to analyze the levels of striatal DA/DOPAC starting from day 1 following MPTP treatment.
6.) Discussion: the authors nicely demonstrate that there are differences in the behavior between controls and MPTP treated mice and between male and female mice (Figure 3). However, there were no differences in the loss of DA between the two genders following MPTP. How do the authors explain the behavioral differences between the two gender groups although there was no dopamine tissue recovery or differential DA loss.

---

## Round 0.2 · accepted · Accept

· Academic Editor

Accept

Please take care of the note from the reviewer, and feel free to include that discussion at proof stage.

Reviewer 2 ·

Basic reporting

nothing to add

Experimental design

Nothing to add

Validity of the findings

Nothing to add

Comments for the author

The authors have adequately responded to the concerns of this reviewer. One small comment. The authors claim, with no evidence provided, that in their acute model, the inflammatory process is complete by the time the mice are tested behaviorally. This claim is most likely incorrect. They should have tested that in their model but based on the very small number of mice used for this entire study, which again should be significantly increased in future studies since their explanation for why such a small number of mice were used was far from adequate, this was not possible. Based on the experience of this reviewer, we have noted continued inflammation at least a month following the last dose of chronic MPTP treatment. This reviewer realizes that there is a difference between acute and chronic MPTP, but the authors need to carry out studies looking at the various inflammatory markers if they are going to continue to make such claims regarding their model.